# Differences in metalloproteinases and their tissue inhibitors in the cerebrospinal fluid are associated with delirium
Mari Aksnes [1] ✉, Mari Haavig Schibstad[2], Farrukh Abbas Chaudhry[3], Bjørn Erik Neerland[4],
Gideon Caplan[5,6], Ingvild Saltvedt[7,8], Rannveig S. Eldholm [7,8], Marius Myrstad[9], Trine Holt Edwin [10],
Karin Persson[10,11], Ane-Victoria Idland[4,12], Christian Thomas Pollmann[13], Roy Bjørkholt Olsen[14],
Torgeir Bruun Wyller[1,4], Henrik Zetterberg [15,16,17,18,19,20], Emma Cunningham [21] & Leiv Otto Watne [1,4,6,22]

## Abstract

**Background** The aetiology of delirium is not known, but pre-existing cognitive impairment is a predisposing factor. Here we explore the associations between delirium and cerebrospinal fluid (CSF) levels of matrix metalloproteinases (MMPs) and their tissue inhibitors (TIMPs), proteins with important roles in both acute injury and chronic neurodegeneration.

**Methods** Using a 13-plex Discovery Assay®, we quantified CSF levels of 9 MMPs and 4 TIMPs in 280 hip fracture patients (140 with delirium), 107 cognitively unimpaired individuals, and 111 patients with Alzheimer's disease dementia. The two delirium-free control groups without acute trauma were included to unravel the effects of acute trauma (hip fracture), dementia, and delirium.

**Results** Here we show that delirium is associated with higher levels of MMP-2, MMP-3, MMP-10, TIMP-1, and TIMP-2; a trend suggests lower levels of TIMP-4 are also associated with delirium. Most delirium patients had pre-existing dementia and low TIMP-4 is the only marker associated with delirium in adjusted analyses. MMP-2, MMP-12, and TIMP-1 levels are clearly higher in the hip fracture patients than in both control groups and several other MMP/TIMPs are impacted by acute trauma or dementia status.

**Conclusions** Several CSF MMP/TIMPs are significantly associated with delirium in hip fracture patients, but alterations in most of these MMP/TIMPs could likely be explained by acute trauma and/or pre-fracture dementia. Low levels of TIMP-4 appear to be directly associated with delirium, and the role of this marker in delirium pathophysiology should be further explored.

## Plain language summary

Delirium is a syndrome in which there are substantial changes in a person's ability to focus, understand, or pay attention to events. Delirium often occurs in response to sudden trauma and is more common in persons with pre-existing cognitive impairment. What happens in the brain during delirium is not well understood. To learn more, we have studied whether markers in the cerebrospinal fluid were altered in people with delirium compared to people without delirium. To understand differences specifically caused by delirium, we included two control groups without acute trauma, one with cognitively healthy participants and one with dementia patients. We found several markers altered in people with delirium, with most of the markers similarly altered in people with cognitive impairment due to dementia. One marker was directly linked to delirium and could potentially shed light on the brain processes that cause the syndrome.

Delirium is characterised by acute, temporary disturbances in attention and cognition[1]. This severe neuropsychiatric syndrome is a common complication of acute illness in hospitalised patients and a strong predictor of future cognitive decline and mortality[2,3]. Older age and cognitive impairment are key predisposing factors for delirium[1,4]. The aetiology of delirium is poorly understood, but it is proposed that delirium occurs when alterations in distinct neurobiological mechanisms increase the brain's vulnerability to acute triggers (e.g., surgery)[1,5].

Matrix metalloproteinases (MMPs) constitute a family of proteinases with central roles in extracellular matrix remodelling, acute and chronic neuroinflammation, and blood-brain barrier (BBB) permeability[6,7]. MMPs

cleave several different substrates including extracellular matrix components, signalling molecules, and inflammatory mediators such as cytokines. Due to their many proteolytic substrates, high MMP activity can be detrimental and is therefore tightly regulated by tissue inhibitors of metalloproteinases (TIMPs)[8]. Expression of different TIMPs has a degree of organ specificity, and TIMP-4 exhibits greater specificity for the brain[9]. Altered MMP activity has been linked to both progressive neurodegenerative diseases such as Alzheimer's disease (AD) and acute neurological injury such as ischemic stroke[10,11]. In aged mice and rats, peripheral surgery appears to induce BBB dysfunction and elevate hippocampal expression of MMP-2 and MMP-9[12-14]. During experimentally induced white matter pathology

---

levels of TIMP-4 are supressed; this may be important for delirium given evidence of altered white matter connectivity on MRI studies and elevated levels of neurofilament light in blood and cerebrospinal fluid (CSF) after delirium[15–17]. Lower plasma levels of MMP-9 have been associated with delirium in critically ill patients[18]. Associations with other MMPs and their inhibitors have not been explored and the understanding of how different MMPs and TIMPs might interact in delirium is limited. Furthermore, the association between CSF MMP and TIMP levels and delirium has not been explored.

The primary aim of this study was to determine whether delirium is associated with alterations in CSF MMP and TIMP levels. We investigated thirteen MMPs and TIMPs in acutely hospitalised patients with hip fracture, with or without delirium and/or dementia. Further, secondary aims were to investigate whether CSF MMPs and TIMPs were connected to any specific clinical aspect of delirium (e.g., incident or prevalent delirium, biomarkers of neuronal injury, mortality) and to disentangle the effects of acute trauma, delirium, and dementia by comparison with two delirium-free control groups without acute trauma (a group of cognitively unimpaired (CU) controls and a group of persons with AD dementia). We show that while several MMPs and TIMPs are altered in delirium, only TIMP-4 is decreased independent of pre-existing cognitive impairment.

## Methods
### Cohorts
This was a multicentre observational study including patients from four hospitals in the Oslo Region, Norway and one hospital in Trondheim, Norway. Patients were included between 2009 and 2019, and CSF samples were available from 280 hip fracture patients (140 with delirium), 107 CU controls, and 111 patients with AD dementia.

**Hip fracture cohort.** The hip fracture patients were included in a multicentre study run at Oslo University Hospital, Diakonhjemmet Hospital, Akershus University Hospital, and Bærum Hospital between 2016 and 2019. All patients admitted for surgical repair of their hip fracture in spinal anaesthesia were eligible for inclusion; informed consent was obtained from all patients or, in the presence of cognitive impairment, from the next of kin. Delirium was assessed according to the DSM-5 criteria based on a standardised procedure. In short, delirium was assessed bedside daily, with tests of cognition and attention[19–21] in all patients until the fifth postoperative day or until discharge in patients with delirium. Patients with delirium were further divided into prevalent delirium (those with delirium at the time of CSF sampling) and incident delirium (those without delirium at the time of CSF sampling who later developed it). In patients not fulfilling all criteria for delirium, subsyndromal delirium was defined as evidence of cognitive change together with any of the following: altered arousal; attentional deficits; other cognitive change; or delusions or hallucinations[22]. Study nurses, trained in delirium assessment by the study physician (LOW), performed all assessments. Two experienced delirium researchers (LOW and BEN) independently assessed all available information for each patient to decide whether the DSM-5 criteria for delirium were fulfilled or not. The interrater agreement upon delirium diagnosis was excellent (kappa 0.97), with disagreements resolved through discussion. Pre-fracture cognitive status was assessed with the Informant Questionnaire on Cognitive Decline in the Elderly (IQCODE); scores ≥ 3.44 were considered to indicate dementia[23]. In the case of missing IQCODE scores ($n = 21$), pre-fracture dementia status was established retrospectively using hospital records. The preoperative American Society of Anesthesiologists (ASA) physical status classification was used as a measure of medical comorbidities[24].

**Cognitively unimpaired control group (CU group).** Cognitively unimpaired individuals aged 65 years or older were recruited from the Cognorm study of patients undergoing elective gynaecological, orthopaedic or urological surgery in spinal anaesthesia at Oslo University

Hospital or Diakonhjemmet Hospital between 2012 and 2013[25]. All patients provided informed consent. The CU group was tested at baseline and at yearly follow-up visits with a standardised battery of cognitive tests[25], and only individuals considered cognitively unimpaired at baseline in line with criteria employed in Knapskog et al.[26] were included. All CU controls were free from delirium at the time of CSF sampling. Patient journals were examined to evaluate incident delirium in the CU group; two patients were determined to have post-operative delirium.

**Alzheimer's disease dementia control group (AD group).** Dementia is a primary risk factor for delirium, and as such there is high overlap between patients with dementia and patients who develop delirium[4]. To disentangle these effects in our study, we included an AD dementia control group (AD group). These patients were assessed at two Norwegian outpatient memory clinics, Oslo University Hospital, and St. Olav University Hospital, and included in the Norwegian Registry of Persons Assessed for Cognitive Symptoms (NorCog) between 2009 and 2018; all patients provided informed consent. Patients were assessed according to a standardised research protocol by experienced memory clinic physicians[27]. All patients met the criteria for probable or possible dementia due to AD[28] and had pathological levels of the AD CSF biomarkers, CSF amyloid-$\beta_{42}$ (A$\beta_{42}$) and phosphorylated tau$_{181}$ (p-tau$_{181}$)[29]. CSF sampling was conducted as part of the diagnostic procedure and was not performed in the presence of precipitating factors for delirium such as sepsis, hip fracture, or other acute illness; all patients in the AD group were free from delirium at the time of CSF sampling and no anaesthetic agent was administered before lumbar puncture.

### CSF sampling and biochemical analysis
In the hip fracture patients and CU group, CSF was collected at the onset of spinal anaesthesia, before anaesthetic agents were administered. In the AD group, CSF was collected as part of the clinical work up for assessment of potential dementia. In all patients, CSF was collected in sterile polypropylene tubes and centrifuged for 10 minutes at 2000 G. The samples were aliquoted into 0.5 ml polypropylene tubes and stored at −80 ℃. All CSF samples were sent on dry ice for biochemical analyses to the Eve Technologies laboratory (Calgary, Canada) where MMPs and TIMPs were analysed using a 13-plex Discovery Assay® on a Luminex® xMAP® instrument. The assay simultaneously measures MMP-1, MMP-2, MMP-3, MMP-7, MMP-8, MMP-9, MMP-10, MMP-12, MMP-13, TIMP-1, TIMP2-, TIMP-3 and TIMP-4 in a single microwell. Samples were measured in duplicate. Eve Technologies regularly performs quality controls to ensure that inter-assay variability falls within the range of 5–20%. CSF A$\beta_{42}$ and p-tau$_{181}$ concentrations for the hip fracture cohort were measured using INNOTEST enzyme-linked immunosorbent assays (Fujirebio) at Sahlgrenska University Hospital (Mölndal, Sweden).

### Statistics and reproducibility
Statistical analyses were performed in STATA 16.1 and data visualisations were created in R4.1.1 using RStudio. Categorical variables were compared using the $\chi^2$ test. Continuous variables were compared using the Mann-Whitney U non-parametric test, as several variables were not normally distributed. We investigated correlations between variables using Spearman's rho correlation. All reported *P*-values are two-sided and due to multiple comparisons only *P*-values less than <0.01 were considered statistically significant.

To investigate the effects of MMPs/TIMPs on delirium we performed univariable and multivariable logistic regression analyses in the hip fracture cohort with delirium as the dependent variable and each measured MMP/TIMP as the independent variable. Moreover, within the delirium patients, we performed multivariate logistic regression analyses for each measured MMP/TIMP with specific features of delirium (presence/absence of hallucinations, illusions, and motoric restlessness) as the dependent variable. To explore the association between the MMPs/TIMPs and 1-year mortality following hip fracture, we performed survival analysis using Cox regression

in the hip fracture patients, censored 365 days after admission to the hospital. All multivariate analyses (logistic and Cox regression) were controlled for sex, age, and dementia status (IQCODE ≥ 3.44 = dementia, IQCODE < 3.44 = no dementia) as these factors are known to influence CSF biomarkers, delirium risk, or both. The survival analysis was adjusted for delirium (presence/absence). One hip fracture patient had extreme values (>15 standard deviations above the mean) of several MMPs and was excluded from further analysis.

MMP-1, MMP-7, MMP-8, MMP-9, and MMP-13 were detectable in less than 50% of samples and were therefore excluded from further analysis. In a minority of samples, MMP-10 (2.4%) and MMP-12 (26.4%) were below the lower limit of quantification (LLQ); for MMP-10 and MMP-12 values below the LLQ were replaced with values randomly drawn from a uniform distribution with a minimum value of 0 and a maximum value of the respective LLQs. MMP-2, MMP-3, and all TIMPs were detectable in all samples.

### Reporting summary

Further information on research design is available in the Nature Portfolio Reporting Summary linked to this article.

## Results

### MMP and TIMP levels in the hip fracture patients

**The effect of delirium status on MMP/TIMP levels.** The characteristics of the hip fracture patients stratified by delirium status are presented in Table 1. The patients with delirium were significantly older (median age 87 years) than the patients without delirium (median age 77 years, $P = 3.7*10^{-12}$). The delirium patients also had significantly higher scores on the IQCODE ($P = 2.0*10^{-15}$) and more severe medical comorbidities with 65.5% ASA-score III-IV versus 38.8% in the patients without delirium, $P = 1.4*10^{-14}$.

Several MMPs and TIMPs differed significantly between the no delirium and delirium groups, see Table 1. MMP-2, MMP-3, MMP-10, TIMP-1, and TIMP-2 were higher in the delirium group, and there was a trend ($P = 0.01$) suggesting lower TIMP-4 in patients with delirium compared to patients without delirium. The distribution of CSF MMP and TIMP levels in the delirium and non-delirium group are presented in Fig. 1.

Within the delirium group, there were no differences in MMP or TIMP levels between patients with the incident ($n = 73$) and prevalent delirium ($n = 67$, all $P > 0.05$). Patients with subsyndromal delirium ($n = 21$) were

included in the no-delirium group; the results were not impacted by the exclusion of these patients in sensitivity analyses.

**The effect of dementia status on MMP and TIMP levels.** The characteristics of the hip fracture patients stratified by dementia status are presented in Supplementary Table 1. The dementia group was significantly older (median age 86 years) than the no dementia group (median age 79.5 years, $P = 9.4*10^{-6}$). Delirium occurred more commonly in the dementia patients (83.9%) than in the non-dementia patients (29.3%, $P = 5.2*10^{-22}$). Compared to hip fracture patients without dementia, the dementia group had higher levels of MMP-10 (median 33.7 pg/mL vs. 20.4 pg/mL, $P = 9.4*10^{-6}$).

**The interplay of dementia and delirium.** As there is a large overlap between hip fracture patients with dementia and with delirium, we further investigated MMP and TIMP levels by stratifying patients by dementia and delirium status. Patient characteristics stratified by dementia and delirium status are presented in Supplementary Table 2.

In patients without dementia, those with delirium ($n = 41$) had significantly higher levels of MMP-3, MMP-10, and TIMP-2 compared to those without ($n = 120$), see Supplementary Table 3. In the patients with dementia, there were no significant differences in MMP and TIMP levels associated with delirium status.

There were no significant differences between patients with incident delirium compared to patients with prevalent delirium, regardless of dementia status, see Supplementary Table 3.

**MMP and TIMP levels as predictors of delirium.** In univariate logistic regressions, higher levels of MMP-2, TIMP-1, and TIMP-2 significantly increased the odds of delirium, see Table 2. In analyses adjusted for sex, age, and cognitive impairment, low TIMP-4 was the only MMP/TIMP significantly associated with delirium. In patients with delirium, none of these markers were associated with increased odds of specific features such as hallucinations ($n = 27$ patients, 22.7%, 21 missing), illusions ($n = 24$ patients, 19.8%, 19 missing) or motoric restlessness ($n = 66$ patients, 51.6%, 12 missing) in multivariate logistic regression analyses, see Supplementary Table 4.

**MMP and TIMP associations with 1-year mortality following hip fracture.** Among the 279 hip fracture patients, there were 59 (21%)

## Table 1 | Characteristics of the hip fracture cohort stratified by delirium-status

|  | All | No delirium | Delirium | **P** |
|---|---|---|---|---|
| *N* | 279 | 139 | 140 |  |
| Age | 84 (74; 89) | 77 (69; 86) | 87 (81; 91) | $3.7*10^{-12}$ |
| Female sex, *n* (%) | 192 (68.8) | 99 (71.2) | 93 (66.4) | 0.37 |
| Dementia, *n* (%) | 118 (42.1) | 19 (16.1) | 99 (83.9) | $5.2*10^{-22}$ |
| IQCODE | 3.3 (3.0; 4.2) | 3.1 (3.0; 3.3) | 3.9 (3.4; 4.8) | $2.0*10^{-15}$ |
| ASA III-IV, *n* (%) | 145 (52.0) | 54 (38.8) | 91 (65.0) | $1.4*10^{-14}$ |
| MMP-2 (ng/mL) | 59.3 (49.3; 69.4) | 56.8 (47.3; 65.0) | 62.2 (52.5; 73.5) | 0.001 |
| MMP-3 (pg/mL) | 227 (154; 335) | 205 (147; 299) | 251 (177; 360) | 0.008 |
| MMP-10 (pg/mL) | 24.2 (13.5; 41.3) | 19.2 (10.3; 31.8) | 32.4 (17.9; 45.7) | $4.7*10^{-6}$ |
| MMP-12 (pg/mL) | 3.6 (0.4; 6.7) | 3.6 (0.4; 7.8) | 3.6 (0.4; 6.7) | 0.59 |
| TIMP-1 (ng/mL) | 87.3 (66.3; 110.0) | 82.3 (60.5; 103.3) | 92.1 (72.7; 121.0) | 0.001 |
| TIMP-2 (ng/mL) | 80.4 (69.0; 93.7) | 76.2 (66.1; 87.3) | 85.7 (71.1; 99.3) | $1.2*10^{-4}$ |
| TIMP-3 (ng/mL) | 15.7 (14.4; 17.0) | 15.6 (14.1; 16.7) | 15.9 (14.4; 17.4) | 0.03 |
| TIMP-4 (ng/mL) | 1.6 (1.3; 1.9) | 1.6 (1.4; 2.0) | 1.5 (1.2; 1.9) | 0.01 |

Data is presented as median (quartile 1; quartile 3) unless otherwise indicated. P value is for comparison between patients with and without delirium using Mann Whitney U test (continuous variables) or χ²-test (categorical variables).
*ASA* American Society of Anesthesiologists physical status classification, *IQCODE* Informant Questionnaire on Cognitive Decline in the Elderly, *MMP* matrix metalloproteinase, *TIMP* tissue inhibitor of matrix metalloproteinase.

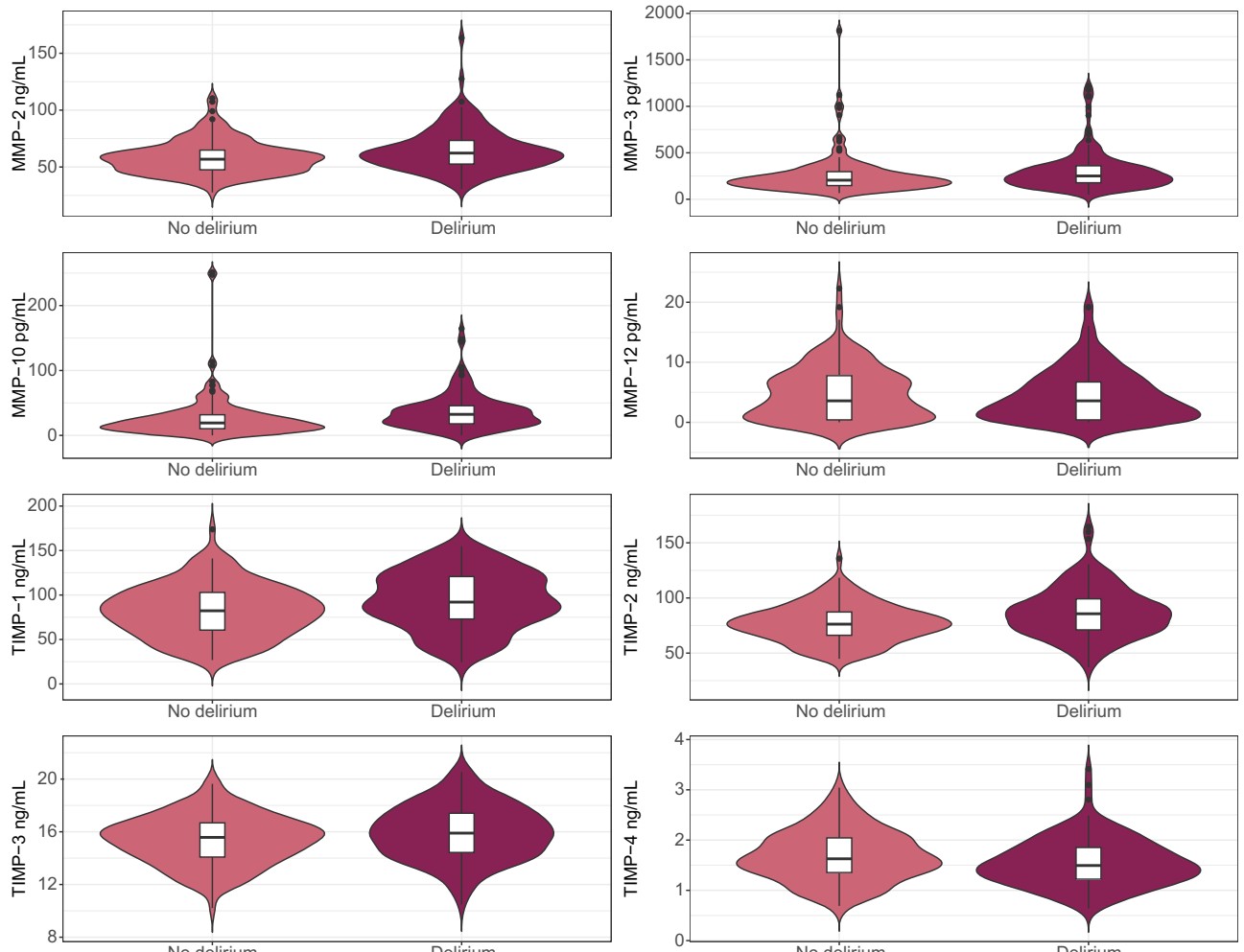

**Fig. 1 |** Distributions of cerebrospinal fluid MMP and TIMP levels across the hip fracture patients with no delirium (*n* = 139) and with delirium (*n* = 140). The violin plots show the entire distribution of each marker, whilst the inserted box plot shows the median (middle line), 1st quartile (lower box limit) and 3rd quartile (upper box limit). Black lines show the minimum (minimum data value − 1.5*interquartile range) and maximum (maximum data value + 1.5* interquartile range). Black circles show outliers. *MMP* matrix metalloproteinase, *TIMP* tissue inhibitor of matrix metalloproteinase.

### Table 2 | Univariate and adjusted logistic regression models predicting odds of delirium in the hip fracture cohort (*n* = 279)

|  | Univariate Odds ratio (95% CI) | *P* | Adjusted* Odds ratio (95% CI) | *P* |
|---|---|---|---|---|
| MMP-2 | 1.025 (1.009; 1.041) | 0.002 | 1.006 (0.988; 1.025) | 0.51 |
| MMP-3 | 1.001 (0.999; 1.002) | 0.11 | 1.000 (0.999; 1.001) | 0.98 |
| MMP-10 | 1.011 (1.002; 1.021) | 0.02 | 1.002 (0.993; 1.011) | 0.65 |
| MMP-12 | 0.977 (0.924; 1.032) | 0.40 | 0.948 (0.883; 1.016) | 0.13 |
| TIMP-1 | 1.013 (1.005; 1.021) | 0.001 | 1.003 (0.992; 1.014) | 0.59 |
| TIMP-2 | 1.026 (1.013; 1.039) | $8.5*10^{-5}$ | 1.015 (0.998; 1.031) | 0.08 |
| TIMP-3 | 1.164 (1.029; 1.318) | 0.02 | 1.020 (0.861; 1.205) | 0.82 |
| TIMP-4 | 0.550 (0.335; 0.902) | 0.02 | 0.424 (0.224; 0.802) | 0.008 |

*Adjusted for sex, age, and presence/absence of dementia (IQCODE ≥ 3.44 = dementia (*n* = 118), IQCODE < 3.44 = no dementia (*n* = 161). In all models, higher age and the presence of dementia were significant predictors of delirium.
*CI* confidence interval, *IQCODE* Informant Questionnaire on Cognitive Decline in the Elderly, *MMP* matrix metalloproteinase, *TIMP* tissue inhibitor of matrix metalloproteinase

deaths in the first year following surgery; 52 (37%) in the delirium group, and 7 in the no-delirium group (5%). In univariate analyses, several MMPs and TIMPs were associated with increased mortality; however, no MMPs or TIMPs were associated with mortality in analyses adjusted for age, sex, delirium, and pre-existing dementia, see Supplementary Table 5.

### Cohort comparisons

We compared the levels of CSF MMP and TIMP in the hip fracture patients with two control groups, the CU group and the AD group; these control groups permitted us to explore how CSF MMP/TIMPs are affected by acute trauma (hip fracture group vs. groups with no acute illness), dementia, and

**Table 3 | Characteristics of the cognitively unimpaired group, the Alzheimer's disease group and the hip fracture patients**

|  | 1. CU group | 2. AD group | 3. Hip fracture | P 1. vs 3. | P 2. vs 3 |
|---|---|---|---|---|---|
| N | 107 | 111 | 279[a] |  |  |
| Age | 71 (67; 76) | 71 (66; 75) | 84 (74; 89) | $1.4*10^{-18}$ | $2.5*10^{-24}$ |
| Female sex | 48 (44.9) | 68 (68.3) | 192 (68.8) | $1.4*10^{-5}$ | 0.15 |
| Delirium | 2 (1.9) | 0 (0) | 140 (50.0) |  |  |
| Dementia[b] | 0 (0) | 111 (100.0) | 118 (42.1) |  |  |
| IQCODE | 3 (3.0; 3.1) | 3.8 (3.5; 4.1) | 3.3 (3.0; 4.2) | $4.9*10^{-19}$ | $4.8*10^{-6}$ |
| ASA III-IV, n (%) | NA | NA | 145 (52.0) |  |  |

Data are presented as n (%) for female sex, delirium, and dementia and as median (quartile 1; quartile 3) for age and IQCODE. P-values are for Mann Whitney U test (continuous variables) and $\chi^2$-test (categorical variables).
[a]One extreme outlier was excluded.
[b]Dementia in hip fracture cohort defined as IQCODE ≥ 3,44. In the case of missing IQCODE scores (n = 21), dementia status was established retrospectively using hospital records.
AD Alzheimer's disease, ASA American Society of Anesthesiologists physical status classification, CU cognitively unimpaired, IQCODE Informant Questionnaire on Cognitive Decline in the Elderly.

delirium. The cohort characteristics are presented in Table 3. The hip fracture patients were significantly older (median age 84 years) than both the CU group (median age 71 years, $P = 1.4*10^{-18}$) and the AD group (median age 71 years, $P = 2.5*10^{-24}$). There was a higher proportion of women in the hip fracture cohort (68.8%) compared to the CU group (44.9%). In the CU group, two patients were diagnosed with incident delirium following surgery based on journal inspection. Sensitivity analyses excluding these patients did not affect the results.

The distribution of CSF MMP and TIMP levels across the three different cohorts is presented in Fig. 2.

Most of the measured MMP/TIMPs differed significantly between the hip fracture patients and the control groups, suggesting that CSF MMP/TIMP levels are affected by the acute trauma, see Table 4 for all comparisons. The markers that were most clearly elevated in the hip fracture patients compared to the other two groups (that did not experience acute trauma) were MMP-2 (~30% higher), MMP-12 (140% higher), and TIMP-1 (>50% higher).

MMP-3 and MMP-10 were the markers most clearly impacted by dementia. Compared to the CU group, the hip fracture cohort (in which 40% had dementia) had 15% higher levels of MMP-3 and 90% higher levels of MMP-10. For both markers, the levels in the AD group were even higher.

### The effect of sex and age
In the hip fracture cohort, there was no difference in the frequency of delirium across men and women. Men had significantly higher levels of MMP-3, MMP-10, and TIMP-2, see Supplementary Table 6.

In all patients combined, age was moderately correlated with MMP-2 levels (rho = 0.58), TIMP-1 levels (rho = 0.54), and TIMP-2 levels (rho = 0.45). Age was weakly correlated with MMP-3 (rho = 0.22) and MMP-10 (rho = 0.28). In the three cohorts, age appeared most strongly correlated with MMP and TIMP levels in the hip fracture patients, and the pattern of correlation was mostly similar across the CU controls and the hip fracture patients. In the AD group, age was only weakly correlated with MMP-2, TIMP-1 and TIMP-2 levels. Correlations between age and all MMPs/TIMPs across the three cohorts are presented in Supplementary Table 7.

### Associations with biomarkers of Alzheimer's disease pathophysiology
In the hip fracture cohort, several of the measured MMPs and TIMPs were weakly to moderately correlated with CSF p-tau$_{181}$ and NFL, see Supplementary Table 8. The strongest correlations were between MMP-10 and p-tau$_{181}$ (rho = 0.46) and MMP-2 and NFL (rho = 0.64). Notably only TIMP-4 was moderately correlated with CSF Aβ$_{42}$ (rho = 0.41).

## Discussion
We have quantified the levels of several MMPs and TIMPs in the CSF of patients with delirium. In a large cohort of hip fracture patients, we found that nearly all measured MMPs and TIMPs differed significantly between patients with and without delirium; in delirium patients, the levels of MMP-2, MMP-3, MMP-10, TIMP-1, and TIMP-2 were higher, and there was a trend suggesting TIMP-4 was lower. However, most of the associations between delirium and MMPs/TIMPs could likely be explained by the acute trauma of the hip fracture and/or pre-fracture dementia. In adjusted analyses, low TIMP-4 was the only marker that significantly increased the odds of delirium. However, neither TIMP-4 nor any of the measured MMP/TIMPs were associated with 1-year mortality following hip fracture.

To our knowledge, this is the first study to investigate CSF MMPs and TIMPs in association with delirium. MMPs/TIMPs have several roles in the central nervous system and the periphery, amongst them the regulation of neuroinflammation[30]. Other markers of neuroinflammation such as CSF interleukin-8 have previously been linked to delirium[31]. However, when stratifying patients by dementia status, most of the group differences were attenuated, suggesting that they could be explained by the prevalence of dementia rather than delirium status. This highlights the importance of documenting pre-existing cognitive impairment in delirium biomarker studies and is a major strength of our study. We did not find any differences in CSF MMP/TIMPs between incident versus prevalent delirium.

We included two control groups, one group with cognitively unimpaired individuals and one with AD dementia patients, permitting us to explore the impact of the acute trauma of hip fracture, dementia, and delirium on MMP/TIMP levels. This is relevant as MMP/TIMPs are highly dynamic systems that are both activated during acute illness and impacted by dementia disorders[10]. Similarly, delirium is precipitated by acute illness and dementia is a key predisposing factor[1,4]. It is therefore a key challenge in delirium biomarker studies to disentangle the effects of acute illness, dementia, and delirium.

Most of the measured MMP/TIMPs differed significantly between the hip fracture patients and the control groups, suggesting that CSF MMP/TIMP levels are affected by acute trauma. The markers that were most clearly elevated in the hip fracture patients were MMP-2, MMP-12, and TIMP-1. It is well established that MMPs and TIMPs play important roles in response to acute traumas such as skin wounds, lung injury, and cardiovascular insults[32–34]. For example, MMP-2 expression is quickly increased in response to oxidative stress, and TIMP-1 plays essential roles in heart tissue remodelling[34]. Similarly, MMPs are upregulated in the central nervous system in response to acute insults such as traumatic brain injury, brain haemorrhage, or ischaemic stroke[8]. Increased expression of MMP-12 is seen both after intracerebral haemorrhage and spinal cord injury in animal models[35,36]; this upregulation is thought to have adverse effects. However, little is known about the effect of acute bodily trauma such as a hip fracture on the levels of these markers in the CSF or the central nervous system more broadly. Our results suggest that CSF MMP-2, MMP-12, and TIMP-1 are elevated in the CSF after hip fracture, but further research is needed to determine whether this is due to increased expression by neural and

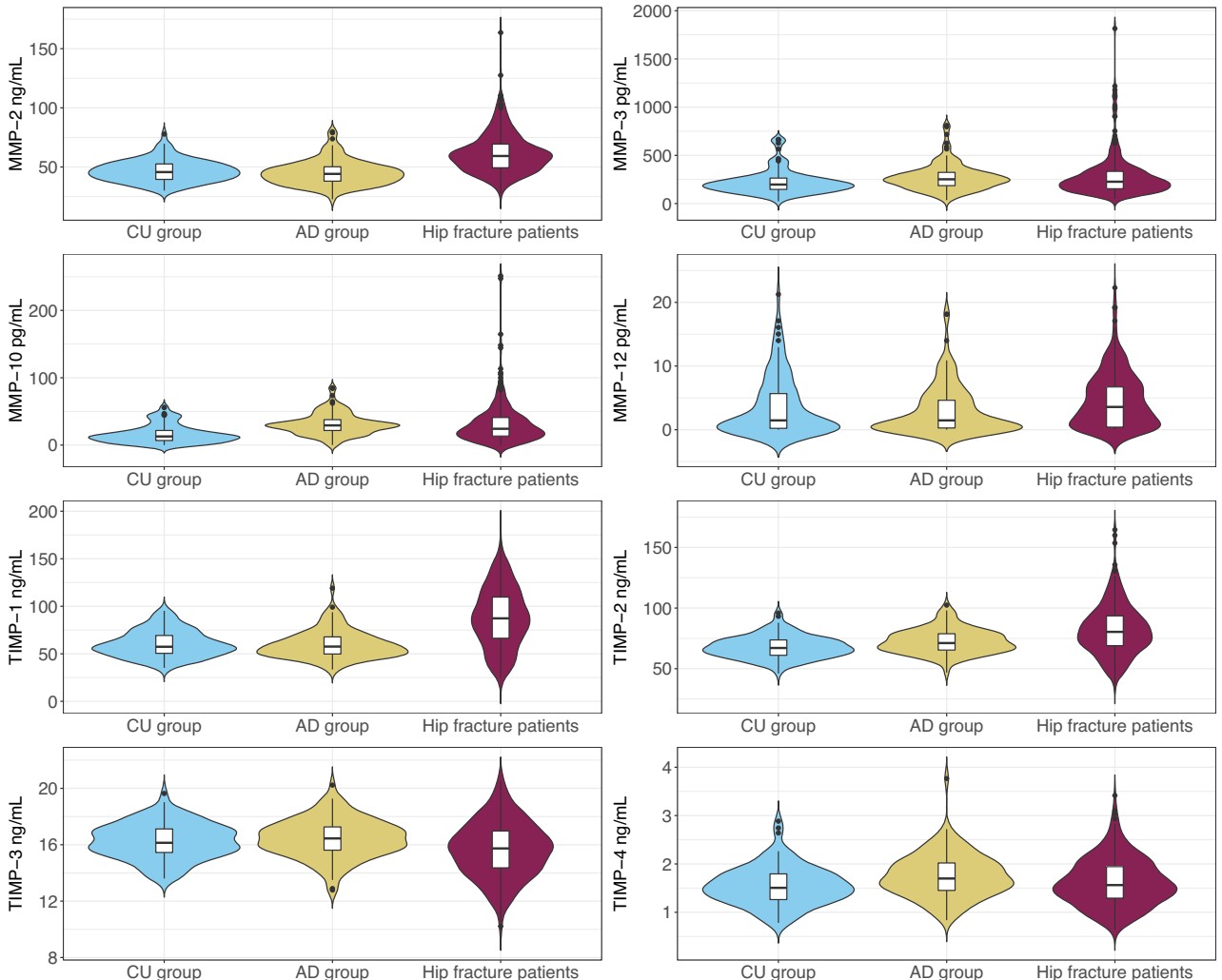

**Fig. 2 |** Distributions of cerebrospinal fluid MMP and TIMP levels across the cognitively unimpaired group ($n = 107$), the Alzheimer's disease group ($n = 111$) and the hip fracture patients ($n = 279$). The violin plots show the entire distribution of each marker, whilst the inserted box plot shows the median (middle line), 1st quartile (lower box limit), and 3rd quartile (upper box limit). Black lines show the minimum (minimum data value − 1.5*interquartile range) and maximum (maximum data value + 1.5* interquartile range). Black circles show outliers. AD Alzheimer's disease, CU cognitively unimpaired, MMP matrix metalloproteinase, TIMP tissue inhibitor of matrix metalloproteinase.

endothelial cells, or increased influx of neutrophils from the periphery[8]. Furthermore, as MMPs have both beneficial and detrimental roles in the central nervous system, it remains to be determined how, if at all, this upregulation contributes to the pathophysiology of delirium.

The levels of MMP-3 and MMP-10 were clearly higher in the hip fracture cohort, in which 40% of patients had dementia, compared to the CU controls, and even higher in the AD dementia cohort. This, along with the modest correlations with age, suggests that elevated levels of MMP-3 and MMP-10 are associated with dementia. This finding is in line with previous research linking increased MMP-10 levels to an increased risk of dementia and accelerated disease progression in neurodegenerative diseases[37,38]. Similarly, CSF levels of MMP-3 have previously been linked to dementia and specifically AD pathology[39–41]. In hip fracture patients, higher levels of MMP-3 and MMP-10 were associated with delirium, but only in those without pre-fracture dementia. This suggests that the expression of MMP-3 and MMP-10 in the CSF after acute injury and in the presence of delirium might be moderated by pre-existing dementia disorders, highlighting the complex interplay between delirium and dementia[4]. Chronic neuroinflammation and changes in the balance between MMPs and TIMPs in dementia disorders might attenuate any response, such as e.g., an increase in MMP-3, to acute

injury in dementia patients. In dementia-free patients, it is possible that higher levels of MMP-3 and MMP-10 indicate reduced brain resilience due to ageing or pre-symptomatic neurodegenerative disease, thus predisposing these patients to delirium.

In analyses adjusted for sex, age, and pre-existing cognitive impairment, only lower levels of TIMP-4 remained a significant predictor for delirium. TIMP-4 is the most recently discovered and least studied member of the TIMP-family[42]. TIMP-4 is generally thought to exert beneficial effects by contributing to extracellular matrix remodelling[43]. CSF levels of TIMP-4 have primarily been studied in patients with bacterial or viral infections; lower levels of CSF TIMP-4 are found in patients with eosinophilic meningitis, syphilis, or HIV-1 and syphilis co-infections[44,45]. One study also found reduced levels of TIMP-4 in the CSF of patients with hydrocephalus[46]. The mechanisms driving the reduced expression of TIMP-4 in these conditions are not established, but it is speculated that alterations in the inflammatory milieu and blood-brain barrier damage are contributing factors[44–46]. In terms of acute effects, there is some indication that TIMP-4 expression should increase in response to trauma, as pericardial TIMP-4 concentrations increase significantly after cardiopulmonary bypass surgery[47]. Interestingly, in our study, TIMP-4 levels did not differ significantly between the CU control group and the delirium patients.

## Table 4 | MMP and TIMP levels in the CU groups, AD group, and hip fracture patients without and with delirium

| | 1. CU group | 2. AD group | Hip fracture patients | | | P | | | |
| --- | --- | --- | --- | --- | --- | --- | --- | --- | --- |
| | | | 3. All | 4. No delirium | 5. Delirium | 1. vs 3. | 2. vs. 3 | 2. vs 4. | 2. vs 5. |
| MMP-2 (ng/mL) | 45.7 (39.5; 52.7) | 44.1 (38.0; 50.2) | 59.3 (49.3; 69.4) | 56.8 (47.3; 65.0) | 62.2 (52.5; 73.5) | $5.4*10^{-21}$ | $1.2*10^{-24}$ | $2.8*10^{-24}$ | $3.2*10^{-15}$ |
| MMP-3 (pg/mL) | 197 (147; 266) | 252 (182; 325) | 227 (154; 335) | 205 (147; 299) | 251 (177; 360) | 0.01 | 0.24 | 0.01 | 0.68 |
| MMP-10 (pg/mL) | 12.8 (6.5; 21.7) | 29.3 (21.7; 38.1) | 24.2 (13.5; 41.3) | 19.2 (10.3; 31.8) | 32.4 (17.9; 45.7) | $2.7*10^{-9}$ | 0.06 | $2.1*10$ | 0.37 |
| MMP-12 (pg/mL) | 1.5 (0.2; 5.7) | 1.5 (0.3; 4.6) | 3.6 (0.4; 6.7) | 3.6 (0.4; 7.8) | 3.6 (0.4; 6.7) | 0.004 | $8.5*10^{-4}$ | 0.003 | 0.003 |
| TIMP-1 (ng/mL) | 57.4 (50.3; 70.1) | 57.6 (49.5; 68.4) | 87.3 (66.3; 110.0) | 82.3 (60.5; 103.3) | 92.1 (72.7; 121.0) | $3.8*10^{-18}$ | $2.6*10^{-19}$ | $1.4*10^{-11}$ | $1.2*10^{-19}$ |
| TIMP-2 (ng/mL) | 67.1 (61.1; 73.7) | 71.1 (65.3; 78.9) | 80.4 (69.0; 93.7) | 76.2 (66.1; 87.3) | 85.7 (71.1; 99.3) | $2.3*10^{-13}$ | $3.7*10^{-6}$ | 0.02 | $3.1*10^{-9}$ |
| TIMP-3 (ng/mL) | 16.1 (15.4; 17.1) | 16.4 (15.6; 17.3) | 15.7 (14.4; 17.0) | 15.6 (14.1; 16.7) | 15.9 (14.4; 17.4) | 0.003 | $5.3*10^{-5}$ | $9.0*10^{-7}$ | 0.03 |
| TIMP-4 (ng/mL) | 1.5 (1.2; 1.8) | 1.7 (1.5; 2.0) | 1.6 (1.3; 1.9) | 1.6 (1.4; 2.0) | 1.5 (1.2; 1.9) | 0.27 | 0.007 | 0.29 | $1.8*10^{-4}$ |

Data is presented as median (quartile 1; quartile 3). P-values are for Mann-Whitney U test.
AD Alzheimer's disease, CU cognitively unimpaired, MMP matrix metalloproteinase, TIMP tissue inhibitor of matrix metalloproteinase.

Moreover, in the hip fracture patients TIMP-4 levels were positively associated with CSF Aβ$_{42}$; low CSF Aβ$_{42}$ is indicative of Alzheimer's disease pathology and known to predict postoperative delirium[48]. We speculate that low TIMP-4 in delirium patients could indicate an insufficient response to acute damage and predispose these patients to delirium, especially in the cases of pre-existing amyloid pathology, but more research is needed to elucidate the contributions of TIMP-4 to delirium pathophysiology.

A major strength of the current study is the large CSF dataset on hip fracture patients with and without delirium. The inclusion of the two control groups, the CU group and the AD group, provides important context for the interpretation of our data. One limitation is the group differences in terms of CSF collection procedures and storage. Importantly, time in room temperature is known to influence CSF MMP levels[49], but CSF was rapidly frozen after collection in all cohorts. However, biobank storage time differed across the three cohorts and may have influenced the results as the effects of long-term storage on CSF MMP/TIMP levels are unknown. Delirium was assessed daily by trained investigators using validated instruments, and the diagnostic algorithm is documented in detail[50]. Furthermore, we have included information on dementia status for all the hip fracture patients, which is essential as dementia is associated with both delirium[4] and MMPs/TIMPs[10,51,52]. However, due to the patients' acute admission, dementia status was determined using the IQCODE, which, whilst validated and commonly used[53], is not a substitute for objective cognitive testing or a complete dementia assessment. We have limited information on the underlying causes of dementia in the dementia group, and these patients might suffer from several interacting pathologies. Moreover, it is possible that the non-dementia group also contained people with undiagnosed underlying neurodegenerative diseases. As much of the previous research on MMPs/TIMPs investigated circulating levels of these markers, it would have strengthened our study to include paired CSF and blood samples.

In conclusion, CSF levels of MMP/TIMPs are altered in hip fracture patients with delirium compared to patients without delirium. Most of these markers can be linked to the acute insult (i.e., the hip fracture) and/or pre-existing dementia in the delirium patients, but TIMP-4 was directly connected to delirium. This study illustrates the importance of collecting information on dementia and including relevant comparison groups in delirium biomarker studies. Further studies should explore the contribution of this marker to delirium pathophysiology.

## Data availability

The numerical data underlying Figs. 1 and 2 can be found in Supplementary Data 1. The further data that support the findings of this study are not openly available in order to preserve the privacy of individual participants under the European General Data Protection Regulation (GDPR). Data are, however, available from the authors upon reasonable request contingent on approval from the ethical committee REC South East (contact at e-mail post@helseforskning.etikkom.no) and for the Alzheimer's disease patients the Norwegian Registry of Persons Assessed for Cognitive Symptoms (NorCog, contact at e-mail: post@aldringoghelse.no).

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

## Acknowledgements

We thank everyone at Oslo University Hospital, Diakonhjemmet Hospital, Akershus University Hospital, and Bærum Hospital who contributed to the inclusion of patients in the hip fracture cohort. We thank the NorCog registry for access to data and biological samples for the AD group. We acknowledge all the patients who have participated in NorCog, Cognorm, or the hip fracture cohort. This work has been funded by the Norwegian Health Association (#1513, #14845, #25633, #19536), the South-Eastern Norway Regional Health Authorities (#2017095), Vestre Viken Hospital Trust and Wellcome Leap's Dynamic Resilience Program (jointly funded by Temasek Trust) (#104617). The funders had no role in the design of the study, data collection, and analysis or preparation of the manuscript. We thank Dr. Muhammad Umar Sajjad for his valuable contributions to this work.

## Author contributions

M.A., F.A.C., T.B.W. and L.O.W. designed the study. M.H.S., B.E.N., I.S., R.S.E., M.M., T.H.E., K.P., A-V.I, C.T.P., R.B.O. and L.O.W. contributed to data collection. B.E.N. and L.O.W. assessed whether the hip fracture patients fulfilled the DSM-5 criteria for delirium. G.C. and E.C. provided expert advice. H.Z. performed biochemical analysis. M.A. performed the statistical analysis, created the data visualisations, and wrote the manuscript. All authors contributed to the interpretation of the results and the revision of the manuscript.

## Competing interests

The authors declare the following competing interests: KP reports work with Novo Nordisk NN6535-4730 trial outside the submitted work. HZ has served at scientific advisory boards and/or as a consultant for Abbvie, Acumen, Alector, Alzinova, ALZPath, Amylyx, Annexon, Apellis, Artery Therapeutics, AZTherapies, Cognito Therapeutics, CogRx, Denali, Eisai, Merry Life, Nervgen, Novo Nordisk, Optoceutics, Passage Bio, Pinteon Therapeutics, Prothena, Red Abbey Labs, reMYND, Roche, Samumed, Siemens Healthineers, Triplet Therapeutics, and Wave, has given lectures in symposia sponsored by Alzecure, Biogen, Cellectricon, Fujirebio, Lilly, Novo Nordisk, and Roche, and is a co-founder of Brain Biomarker Solutions in Gothenburg AB (BBS), which is a part of the GU Ventures Incubator Program (outside submitted work). All other authors explicitly report no disclosures.

## Ethics approval and consent to participate

The study was conducted in accordance with the Declaration of Helsinki. All data and materials were collected after informed written consent from the patient, or from the next of kin if patients were unable to consent due to cognitive impairment; next of kin have the authority to grant consent in Norway under the Act on Medical and Health Research (ACT 2008-06-20). The Regional Committee for Ethics in Medical Research in Norway has approved the multi-centre study from which hip fracture patients were included (REC Central, #19337); the Cognorm study (REC South East, #12064); and the study of inflammatory biomarkers in the NorCog material (REC South East #6897). The ethical approval encompasses the current study, including analysis of MMP and TIMP biomarkers and comparisons across cohorts.

## Additional information

¹Institute of Clinical Medicine, University of Oslo, Oslo, Norway. ²Department of Geriatric Medicine, Sørlandet Hospital, Arendal, Norway. ³Department of Molecular Medicine, Institute of Basic Medical Sciences, University of Oslo, Oslo, Norway. ⁴Oslo Delirium Research Group, Department of Geriatric Medicine, Oslo University Hospital, Oslo, Norway. ⁵Department of Geriatric Medicine, Prince of Wales Hospital, Sydney, NSW, Australia. ⁶Prince of Wales Clinical School, University of New South Wales, Sydney, NSW, Australia. ⁷Department of Neuromedicine and Movement Science, Norwegian University of Science and Technology, Trondheim, Norway. ⁸Department of Geriatric Medicine, St. Olavs Hospital, Trondheim University Hospital, Trondheim, Norway. ⁹Department of Internal Medicine, Bærum Hospital, Vestre Viken Hospital Trust, Bærum, Norway. ¹⁰Department of Geriatric Medicine, Oslo University Hospital, Oslo, Norway. ¹¹Vestfold Hospital Trust, Norwegian National Centre for Ageing and Health, Tønsberg, Vestfold, Norway. ¹²Department of Anesthesiology, Akershus University Hospital, Lørenskog, Norway. ¹³Department of Orthopedic Surgery, Akershus University Hospital, Oslo, Norway. ¹⁴Department of Anesthesiology and Intensive Care, Sørlandet Hospital, Arendal, Norway. ¹⁵Institute of Neuroscience and Physiology, the Sahlgrenska Academy at University of Gothenburg, Mölndal, Sweden. ¹⁶Clinical Neurochemistry Laboratory, Sahlgrenska University

Hospital, Mölndal, Sweden. [17]Department of Neurodegenerative Disease, UCL Institute of Neurology, London, UK. [18]UK Dementia Research Institute at UCL, London, UK. [19]Hong Kong Center for Neurodegenerative Diseases, Hong Kong, China. [20]Wisconsin Alzheimer's Disease Research Center, University of Wisconsin School of Medicine and Public Health, University of Wisconsin-Madison, Madison, WI, USA. [21]Centre for Public Health, Queen's University Belfast, Belfast, UK. [22]Department of Geriatric Medicine, Akershus University Hospital, Lørenskog, Norway. ✉e-mail: mari.aksnes@medisin.uio.no

