## [Peer Review File · Communications Medicine]

Reviewers' comments:

Reviewer #1 (Remarks to the Author):

General Comments:

I appreciate the opportunity to review this elegantly designed and conducted manuscript titled "Matrix metalloproteinases and their tissue inhibitors in delirium". The authors have addressed an intriguing topic on the role of matrix metalloproteases (MMPs) and their tissue inhibitors (TIMPs) in CSF of patients that developed delirium during postoperative period after a hip fracture surgery and have included two control groups to be able to disentangle the effects of age, previous cognitive status, and acute illness on these proteins and illustrate their changes in delirium. The methodology, design of the study and the statistical analysis are sound and appropriate for the aims of the study. The authors have robustly demonstrated that most of the differential expression of these proteins in delirious patients can be explained by dementia (MMP-10 and MMP-3), effects of acute illness (TIMP-1, TIMP-2, and MMP-12), and age (MMP-2, TIMP-1, and TIMP-2). TIMP-4 is found to be significantly lower in delirium group and close to its levels in cognitively unimpaired group after adjustment for age, sex, and cognitive status.

Minor Comments:

- I might be wrong but in the supplementary table 1 in the Dementia line and Dementia column the number and percentage are written down as 100(100). Please check if it needs to be 118 (100).
- I suggest providing violin plots that divide the hip fracture group into delirious vs. not delirious subsections instead of/in addition to merging them all together since delirium is the ultimate concept of interest in relation to MMP/TIMP concentrations.

Reviewer #2 (Remarks to the Author):

In this manuscript, Aksnes et al. analyze the levels of 9 MMPs and 4 TIMPs in 240 patients with hip fracture (140 with delirium), 107 CU controls and 111 patients with AD dementia. The authors found that TIM-4 was the only marker associated with delirium.

The cohort included in the study has a large sample size and is well-characterized. The differences in these markers in AD have been previously investigated, but the role in delirium is novel and of interest.

Main comments:

- The authors analyze the data based on the presence/absence of delirium. It would be interesting to also correlate the markers with specific components of delirium (hallucinations, agitation, cognitive measures, etc). This would add more granularity to the results.
- It would be also interesting to correlate the levels of these markers with future outcomes (for example at 6 or 12 months).
- The authors comment on the effect of age. It would be also interesting to investigate the association between MMPs and TIMPs and core AD biomarkers, especially total tau and p-tau.

Reviewer #3 (Remarks to the Author):

In this study, researchers investigated the possible link between delirium and cerebrospinal fluid (CSF) levels of matrix metalloproteinases (MMPs) and their tissue inhibitors (TIMPs), which are important proteins in acute injury and chronic neurodegeneration. They analyzed CSF samples from 280 hip fracture patients, out of which 140 had delirium, 107 were cognitively unimpaired individuals, and 111 had Alzheimer's disease dementia. The study found that delirium in hip fracture patients was connected to elevated levels of MMP-2, MMP-3, MMP-10, TIMP-1, TIMP-2, and TIMP-3, and decreased levels of TIMP-4. The researchers further analyzed the data and discovered that low TIMP-4 was the only marker that was significantly associated with delirium in patients with pre-existing dementia. These findings suggest that certain CSF MMP/TIMPs may be connected to delirium in hip fracture patients, but trauma and pre-existing dementia could also be responsible for most of these changes, with low TIMP-4 being a direct factor. Overall, this study answers important questions and makes use of hard-to-obtain samples from different cohorts. Suggestions to improve the manuscript are below:

Discussion Section:

A minor concern arises regarding the statement, "in multiple neurotransmitter and neuroinflammatory systems increase the brain's," which overlooks various other potential mechanisms of delirium, such as metabolic or mitochondrial insufficiency. It is advisable to adopt a more general approach, referring to alterations in normal biological functioning.

Methods Section:

A major concern arises from the extensive data presented, making it challenging to follow the study. To enhance clarity, the manuscript could focus on postoperative delirium instead of studying both prevalent delirium and incident delirium groups. Given samples were obtained preoperatively, prevalent delirium in this instance is likely secondary to difference etiology (e.g., pain, infection, narcotic, sedative). This focused approach would streamline the paper, reducing the number of tables and confounding variables. Clearly defining a primary outcome related to postoperative delirium and relegating other findings to secondary outcomes would enhance clarity.

A minor concern is the absence of a reference to the manuscript validating the definition of subsyndromal delirium, which should be addressed for completeness.

A moderate concern is raised regarding the assessment of delirium criteria by two investigators, potentially leading to biases in the advent once of the investigators was more vocal. To address this, given a third adjudicator was not employed, a sensitivity analysis excluding cases with disagreements between raters could provide more robust results.

A major concern stems from the use of the cognitively unimpaired group as a control, considering the differences in sample collection years and the lack of assessment for postoperative delirium is a non-trivial confound. This also choice introduces other confounding factors, making result interpretation challenging. This data can be used to study to study prevalent delirium (if delirium was indeed assessed at the time of sample collection), perhaps in another manuscript, instead of incident delirium.

A minor concern pertains to the statement that all patients in the Alzheimer's disease group were free from delirium, lacking an explanation of how this was determined given the nuanced

assessment challenges in patients with dementia.

Statistical Methods:

A major concern arises due to the numerous comparisons and lack of a defined primary outcome, necessitating P-value correction for multiple comparisons as appropriate to minimize false positives.

A moderate concern involves the clarification of how random values for MMP values below the lower limit of detection were chosen. Providing details on the method used for random selection, such as an automated draw or selecting from the bottom 5% left tail of the data distribution, would enhance transparency. A sensitivity analysis with missing data coding would also be valuable.

Results:

A minor concern pertains to the adjusted model, where clarification on how cognitive impairment (dementia as a binary variable vs. IQCODE as a continuous variable) was defined and the rationale for including specific adjustment variables would enhance the understanding of the results presented in the text.

Tables:

A moderate concern is raised regarding missing important demographic information in Table 1, such as anesthetic type, intraoperative medications, and comorbidities. Additionally, specifying the type of delirium studied, whether preoperative or postoperative, would aid in interpreting the data correctly.

Moderate concern also exists for Tables 2 and 3, which could be simplified by focusing on postoperative delirium and running different models for MMPs/TIMPs while adjusting for dementia. Complex data could be relegated to supplemental materials for clarity.

A major concern surrounds Table 4, as it forms the core of the paper. Understanding the underlying data, including total sample size and the inclusion of patients with dementia, is essential. Explicitly adjusting for dementia, rather than IQCODE, may be appropriate here. Providing insights into the approach used for data variables chosen as covariates of interest (data-driven, clinical intuition, or both) would enhance the readers' understanding of the results.

Author responses to reviewer comments:

We thank all the reviewers for taking the time to consider our manuscript and providing thoughtful feedback. Please see below our point-by-point responses to each comment, in blue. Changes made in the manuscript are highlighted in *italics* and all references to line- numbers refer to the revised manuscript with all tracked changes displayed.

Reviewers' comments:

Reviewer #1 (Remarks to the Author):

General Comments:

I appreciate the opportunity to review this elegantly designed and conducted manuscript titled "Matrix metalloproteinases and their tissue inhibitors in delirium". The authors have addressed an intriguing topic on the role of matrix metalloproteases (MMPs) and their tissue inhibitors (TIMPs) in CSF of patients that developed delirium during postoperative period after a hip fracture surgery and have included two control groups to be able to disentangle the effects of age, previous cognitive status, and acute illness on these proteins and illustrate their changes in delirium. The methodology, design of the study and the statistical analysis are sound and appropriate for the aims of the study. The authors have robustly demonstrated that most of the differential expression of these proteins in delirious patients can be explained by dementia (MMP-10 and MMP-3), effects of acute illness (TIMP-1, TIMP-2, and MMP-12), and age (MMP-2, TIMP-1, and TIMP-2). TIMP-4 is found to be significantly lower in delirium group and close to its levels in cognitively unimpaired group after adjustment for age, sex, and cognitive status.

Response: We thank the reviewer for this encouraging acknowledgment of our work. Your suggestions and corrections have helped us improve the quality of the paper.

Minor Comments:

- **(1)** I might be wrong but in the supplementary table 1 in the Dementia line and Dementia column the number and percentage are written down as 100(100). Please check if it needs to be 118 (100)

Response 1: Supplementary Table 1 has been updated so that it now reads 118 (100), we thank the reviewer for pointing out this error.

- **(2)** I suggest providing violin plots that divide the hip fracture group into delirious vs. not delirious subsections instead of/in addition to merging them all together since delirium is the ultimate concept of interest in relation to MMP/TIMP concentrations.

Response 2: Thank you for this suggestion. In the revised manuscript, we have reorganised the Results-section to focus more clearly on delirium. Per the reviewer's comment we have included an additional violin plot showing the distribution of MMP and TIMP across the delirium versus no-delirium group; Figure 1 in the revised manuscript. The original violin plot is Figure 2 in the revised manuscript.

Reviewer #2 (Remarks to the Author):

In this manuscript, Aksnes et al. analyze the levels of 9 MMPs and 4 TIMPs in 240 patients with hip fracture (140 with delirium), 107 CU controls and 111 patients with AD dementia. The authors found that TIM-4 was the only marker associated with delirium.

The cohort included in the study has a large sample size and is well-characterized. The differences in these markers in AD have been previously investigated, but the role in delirium is novel and of interest.

Response: We thank the reviewer for taking the time to evaluate our manuscript and for the thoughtful comments provided.

Main comments:

- **(1)** The authors analyze the data based on the presence/absence of delirium. It would be interesting to also correlate the markers with specific components of delirium (hallucinations, agitation, cognitive measures, etc). This would add more granularity to the results.

Response 1: We thank the reviewer for this suggestion and agree that it would be interesting to explore associations between the measured MMP/TIMPs and specific features of delirium. In our cohort, we have collected information on hallucinations, illusions and motoric restlessness. In logistic regression analyses in the delirium cohort using presence/absence of these features as the dependent variables (adjusted for sex, age and pre-existing cognitive impairment) we found no significant associations between these features and the measured MMP/TIMPs. This is described in the manuscript:

Methods (lines 189-191): *“Moreover, within the delirium patients we performed multivariate logistic regression analyses for each measured MMP/TIMP with specific features of delirium (presence/absence of hallucinations, illusions and motoric restlessness) as the dependent variable.”*

Results (lines 258-262): *“In patients with delirium, none of these markers were associated increased odds of specific features such as hallucinations (n = 27 patients, 22.7 %, 21 missing), illusions (n = 24 patients, 19.8 %, 19 missing) or motoric restlessness (n = 66 patients, 51.6 %, 12 missing) in multivariate logistic regression analyses (data not shown).”*

- **(2)** It would be also interesting to correlate the levels of these markers with future outcomes (for example at 6 or 12 months).

Response 2: We thank the reviewer for this suggestion and agree that this is an important aspect to investigate. As such, we have investigated the association between these markers and 1-year mortality after hip fracture, please see the following sections in the revised manuscript:

Methods (lines 192-198): *“To explore the association between the MMPs/TIMPs and 1-year mortality following hip fracture, we performed survival analysis using Cox regression in the hip fracture patients, censored 365 days after admission to the hospital. All multivariate analyses (logistic and Cox regression) were controlled for sex, age, and dementia status (IQCODE \geq 3.44 = dementia, IQCODE $<$ 3.44 = no dementia) as these factors are known to influence CSF biomarkers, delirium risk or both. The survival analysis was adjusted for delirium (presence/absence).”*

Results (lines 264-269): ***MMP and TIMP associations with 1-year mortality following hip fracture***

Among the 279 hip fracture patients, there were 59 (21%) deaths in the first year following surgery; 52 (37%) in the delirium group and 7 in the no-delirium group (5 %). In univariate analyses, several MMPs and TIMPs were associated with increased mortality; however, no MMPs or TIMPs were associated with mortality in analyses adjusted for age, sex, delirium and pre-existing dementia, see Supplementary Table 4.”

Discussion (lines 320-322): *“However, neither TIMP-4 nor any of the measured MMP/TIMPs were associated with 1-year mortality following hip fracture.”*

- **(3)** The authors comment on the effect of age. It would be also interesting to investigate the association between MMPs and TIMPs and core AD biomarkers, especially total tau and p-tau.

Response 3: In line with your suggestion, we have included a section in the Results investigating associations between the measured MMP/TIMPs and biomarkers of AD pathophysiology, namely CSF A β ₄₂, p-tau₁₈₁ and neurofilament light (NFL) in the hip fracture cohort. Results (lines 307-311):

“Associations with biomarkers of Alzheimer’s disease pathophysiology

In the hip fracture cohort, several of the measured MMPs and TIMPs were weakly to moderately correlated with CSF p-tau₁₈₁ and NFL, see Supplementary Table 7. The strongest correlations were between MMP-10 and p-tau₁₈₁ ($\rho = 0.46$) and MMP-2 and NFL ($\rho = 0.64$). Notably only TIMP-4 was moderately correlated with CSF A β ₄₂ ($\rho = 0.41$).”

The full results of the correlation are presented in Supplementary Table 7.

We have also made some minor edits to the Discussion in order to address these results, Discussion (lines 387-392): *“Moreover, in the hip fracture patients TIMP-4 levels were positively associated with CSF A β ₄₂; low CSF A β ₄₂ is indicative of Alzheimer’s disease pathology and known to predict postoperative delirium⁴⁸. We speculate that low TIMP-4 in the delirium*

patients could indicate an insufficient response to acute damage and predispose these patients to delirium, *especially in the cases of pre-existing amyloid pathology*, but more research is needed to elucidate the contributions of TIMP-4 to delirium pathophysiology.”

Reviewer #3 (Remarks to the Author):

In this study, researchers investigated the possible link between delirium and cerebrospinal fluid (CSF) levels of matrix metalloproteinases (MMPs) and their tissue inhibitors (TIMPs), which are important proteins in acute injury and chronic neurodegeneration. They analyzed CSF samples from 280 hip fracture patients, out of which 140 had delirium, 107 were cognitively unimpaired individuals, and 111 had Alzheimer's disease dementia. The study found that delirium in hip fracture patients was connected to elevated levels of MMP-2, MMP-3, MMP-10, TIMP-1, TIMP-2, and TIMP-3, and decreased levels of TIMP-4. The researchers further analyzed the data and discovered that low TIMP-4 was the only marker that was significantly associated with delirium in patients with pre-existing dementia. These findings suggest that certain CSF MMP/TIMPs may be connected to delirium in hip fracture patients, but trauma and pre-existing dementia could also be responsible for most of these changes, with low TIMP-4 being a direct factor. Overall, this study answers important questions and makes use of hard-to-obtain samples from different cohorts. Suggestions to improve the manuscript are below:

Response: We thank the reviewer for taking the time to evaluate our manuscript and providing detailed suggestions for improvement. Please see our point-by-point responses below.

Discussion Section:

(1) A minor concern arises regarding the statement, "in multiple neurotransmitter and neuroinflammatory systems increase the brain's," which overlooks various other potential mechanisms of delirium, such as metabolic or mitochondrial insufficiency. It is advisable to adopt a more general approach, referring to alterations in normal biological functioning.

Response 1: We agree with the reviewer that this statement omits several potential contributors to delirium aetiology and have revised the sentence (lines 79-82): “The aetiology of delirium is poorly understood, *but it is proposed that delirium occurs when alterations in distinct neurobiological mechanisms* increase the brain’s vulnerability to acute triggers (e.g. surgery)^{1, 5}.”

Methods Section:

(2) A major concern arises from the extensive data presented, making it challenging to follow the study. To enhance clarity, the manuscript could focus on postoperative delirium instead of studying both prevalent delirium and incident delirium groups. Given samples were obtained

preoperatively, prevalent delirium in this instance is likely secondary to difference etiology (e.g., pain, infection, narcotic, sedative). This focused approach would streamline the paper, reducing the number of tables and confounding variables. Clearly defining a primary outcome related to postoperative delirium and relegating other findings to secondary outcomes would enhance clarity.

Response 2. We acknowledge that the manuscript presents extensive data and that reducing the number of complicated tables and groups included would improve the flow and ease interpretation of the data.

While we agree with the reviewer that focusing on postoperative delirium would streamline our paper, our opinion is that excluding these patients would reduce the quality of our manuscript. As it is well established that cognitive impairment is a significant predictor for pre-operative or prevalent delirium, excluding the patients with prevalent delirium from the paper also confers excluding a significant proportion of those with pre-fracture dementia (45 %) from the paper. This in turn may lead to the exclusion of those patients most prone to develop delirium and result in an underestimation of the importance of preoperative cognitive impairment as a risk factor for delirium (both prevalent and incident delirium), see for example Fong and Inouye (2022, <https://doi.org/10.1038/s41582-022-00698-7>). We believe it is important to include these patients to increase knowledge about risk factors for the development of delirium in this group. Furthermore, while little is currently known about the role of MMP/TIMPs in delirium, it is possible that these concentrations fluctuate during a delirium episode. Whilst it would be optimal to explore this with repeated CSF sampling during a delirium episode, this is ethically and technically challenging; however, including both patients with preoperative delirium (*i.e.* patients with ongoing delirium when CSF was sampled) and postoperative delirium (*i.e.* patients free from delirium when CSF was sampled, but who developed it later) we were able to explore whether some MMP/TIMPs are more closely associated with ongoing delirium (preoperative delirium) while others are more closely associated with the phase where patients demonstrate clinical symptoms of delirium (postoperative delirium).

We have made other efforts to address this concern. Firstly, we have noted the primary objective in the Introduction (lines 102-110):

“The primary aim of this study was to determine whether delirium is associated with alterations in CSF MMP and TIMP levels. We investigated thirteen MMPs and TIMPs in acutely hospitalised patients with hip fracture, with or without delirium and/or dementia. Further, *secondary aims were to investigate whether CSF MMPs and TIMPs were connected to any specific clinical aspect of delirium (e.g. incident or prevalent delirium, biomarkers of neuronal injury, mortality) and to disentangle the effects of acute trauma, delirium and dementia by comparison with two delirium-free control groups without acute trauma (a group of cognitively unimpaired (CU) controls and a group of persons with AD dementia).*”

Furthermore, we have edited the Results section to more clearly focus on the primary outcome (the association between MMP/TIMP levels and delirium) and have moved one table (original Table 3) to the Supplementary Material. We hope the reviewer will agree that this streamlines the manuscript while still including all the important data upon which we build our conclusions.

(3) A minor concern is the absence of a reference to the manuscript validating the definition of subsyndromal delirium, which should be addressed for completeness.

Response 3: To our knowledge, there is no standardized definition of subsyndromal delirium, but the term has been introduced to describe the clinical condition that falls between no symptoms of delirium and delirium as defined by the DSM-5 criteria and it has been argued that inattention should be central to the definition (see Meagher et al., 2014, <https://doi.org/10.1192/bjp.bp.113.139865>). We have included a reference describing the diagnostic criteria used for subsyndromal delirium in more detail: Neerland, B.E., Hov, K.R., Bruun Wyller, V. et al. The protocol of the Oslo Study of Clonidine in Elderly Patients with Delirium; LUCID: a randomised placebo-controlled trial. *BMC Geriatr* 15, 7 (2015). <https://doi.org/10.1186/s12877-015-0006-3>

(4) A moderate concern is raised regarding the assessment of delirium criteria by two investigators, potentially leading to biases in the advent once of the investigators was more vocal. To address this, given a third adjudicator was not employed, a sensitivity analysis excluding cases with disagreements between raters could provide more robust results.

Response 4: We appreciate the reviewer's concern and acknowledge that consulting a third adjudicator in the case of disagreement would limit any bias. However, we would like to point out that the interrater agreement was near perfect with a Cohen's kappa of 0.97, corresponding to disagreement in less than 2 % of cases. As such, we do not believe a sensitivity analysis excluding these few patients would have an impact on the results.

(5) A major concern stems from the use of the cognitively unimpaired group as a control, considering the differences in sample collection years and the lack of assessment for postoperative delirium is a non-trivial confound. This also choice introduces other confounding factors, making result interpretation challenging. This data can be used to study to study prevalent delirium (if delirium was indeed assessed at the time of sample collection), perhaps in another manuscript, instead of incident delirium.

Response 5: We acknowledge the reviewer's concern regarding the cognitively unimpaired group and hope to assuage them with the following comments and revisions:

1. None of the patients in the cognitively unimpaired group had prevalent/preoperative delirium. This would have disqualified them from elective surgery and inclusion in the cognitively unimpaired cohort.

2. Post-operative/incident delirium has been assessed in the cognitively unimpaired cohort based on medical journal notes. Cross-checking with the patients included in the current study, two appear to have experienced post-operative delirium. Exclusion of these patients in sensitivity analysis did not affect the results or conclusions in the manuscript.

We have noted this in the manuscript:

Methods (lines 149-150): *“Patient journals were examined to evaluate incident delirium in the CU group; two patients were determined to have post-operative delirium.”*

Results (lines 278-280): *“In the CU group, two patients were diagnosed with incident delirium following surgery based on journal inspection. Sensitivity analyses excluding these patients did not affect the results.”*

3. The difference in sample collection years is certainly a limitation. We have created scatter plots to visually assess whether there is any clear pattern suggestion that longer freezer durations affect the measured concentrations of MMP/TIMPs, but no clear pattern emerges suggesting that storage time is associated with lower or higher levels of MMP/TIMPs. To our knowledge, there are no publications that have assessed the effect of long-term storage in a biobank on CSF MMP/TIMP levels. We have however noted this limitation in the Discussion (lines 396-401): *“One limitation of the current study is the group differences in terms of CSF collection procedures and storage. Importantly, time in room temperature is known to influence CSF MMP levels⁴⁹, but CSF was rapidly frozen after collection in all cohorts. However, biobank storage time differed across the three cohorts and may have influenced the results as the effects of long-term storage on CSF MMP/TIMP levels is unknown.”*
4. While we acknowledge that the inclusion of a control group introduces confounding factors to the manuscript, we believe this group is essential to demonstrate that many of the apparent differences in MMP/TIMP levels associated with delirium, are indeed better explained by advanced age, dementia and/or acute trauma. The cognitively unimpaired group includes patients without acute trauma and without dementia, whilst all other included patients have the presence of either or both.

(6) A minor concern pertains to the statement that all patients in the Alzheimer's disease group were free from delirium, lacking an explanation of how this was determined given the nuanced assessment challenges in patients with dementia.

Response 6: The patients in the Alzheimer's disease group were examined at the outpatient memory clinic. In cases where patients were referred from inpatient hospital stays, the examination was scheduled at least three months after the hospital stay. In the presence of precipitating factors for delirium such as acute illness, patients are not referred to CSF sampling. To highlight this, we have added the following text to the manuscript, Methods (lines 154-158): *“These patients were assessed at two Norwegian outpatient memory clinics, Oslo University Hospital and St. Olav University Hospital, and included in the Norwegian Registry of Persons*

Assessed for Cognitive Symptoms (NorCog) between 2009 and 2018; *all patients provided informed consent.*”

Methods lines 161-164:

“CSF sampling was conducted as part of the diagnostic procedure and was not performed in the presence of precipitating factors for delirium such as sepsis, hip fracture or other acute illness; all patients in the AD group were free from delirium at the time of CSF sampling and no anaesthetic agent was administered before lumbar puncture.”

Statistical Methods:

(7) A major concern arises due to the numerous comparisons and lack of a defined primary outcome, necessitating P-value correction for multiple comparisons as appropriate to minimize false positives.

Response 7: As noted in our previous response (2), we have attempted to define the primary outcome of the analyses more clearly in the Introduction and Results sections. Moreover, to limit the number of false positives whilst still avoiding the risk of Type II errors, we have adjusted the chosen α from 0.05 to 0.01, thus considering $P < 0.01$ statistically significant. As such, a few findings are no longer considered statistically significant: notably TIMP-3 is not significantly increased in the delirium group vs. the delirium group ($P = 0.03$) and there is only a trend suggesting reduced levels of TIMP-4 ($P = 0.01$). The manuscript has been edited to reflect this, however, the main conclusions of the study are not affected.

(8) A moderate concern involves the clarification of how random values for MMP values below the lower limit of detection were chosen. Providing details on the method used for random selection, such as an automated draw or selecting from the bottom 5% left tail of the data distribution, would enhance transparency. A sensitivity analysis with missing data coding would also be valuable.

Response 8: We have edited the description of the selection of random values for the MMP-10 and MMP-12 values below the lower limit of quantification to enhance transparency, Methods lines (202-206): *“In a minority of samples, MMP-10 (2.4 %) and MMP-12 (26.4 %) were below the lower limit of quantification (LLQ); for MMP-10 and MMP-12 values below the LLQ were replaced with values randomly drawn from a uniform distribution with a minimum value of 0 and a maximum value of the respective LLQs”*

Results:

(9) A minor concern pertains to the adjusted model, where clarification on how cognitive

impairment (dementia as a binary variable vs. IQCODE as a continuous variable) was defined and the rationale for including specific adjustment variables would enhance the understanding of the results presented in the text.

Response 9: We have amended the Methods section to clarify that analyses were adjusted for dementia as a binary variable, with IQCODE ≥ 3.44 being considered dementia and IQCODE < 3.44 being considered no dementia and have added a short statement to justify the choice of covariates of interest, Methods (lines 195-198): “*All multivariate analyses (logistic and Cox regression) were controlled for sex, age, and dementia status (IQCODE ≥ 3.44 = dementia, IQCODE < 3.44 = no dementia) as these factors are known to influence CSF biomarkers, delirium risk or both.*”

Tables:

(10) A moderate concern is raised regarding missing important demographic information in Table 1, such as anesthetic type, intraoperative medications, and comorbidities. **(11)** Additionally, specifying the type of delirium studied, whether preoperative or postoperative, would aid in interpreting the data correctly.

Response 10: To address this comment, we have included the preoperative American Society of Anesthesiologists (ASA) physical status classification in all tables showing demographic information. The ASA can be used as a proxy measure of comorbidities. See Methods (lines 138-140): “*The preoperative American Society of Anesthesiologists (ASA) physical status classification was used as a measure of medical comorbidities²⁴.*”

For both the hip fracture patients and the CU group, the type of anaesthetic was spinal. For the hip fracture cohort, we have clarified this in the Methods (lines 120-122): “*All patients admitted for surgical repair of their hip fracture in spinal anaesthesia were eligible for inclusion; informed consent was obtained from all patients or, in the presence of cognitive impairment, from a proxy*”. For the CU cohort, this is stated in the Methods (lines 142-145): “*Cognitively unimpaired individuals aged 65 years or older were recruited from the Cognorm study of patients undergoing elective gynaecological, orthopaedic or urological surgery in spinal anaesthesia at Oslo University Hospital or Diakonhjemmet Hospital between 2012 and 2013²⁵*”. For the AD patients, no anaesthetic was used during lumbar puncture, this has been specified in the Methods (lines 161-164): “*CSF sampling was conducted as part of the diagnostic procedure and was not performed in the presence of precipitating factors for delirium such as sepsis, hip fracture or other acute illness; all patients in the AD group were free from delirium at the time of CSF sampling and no anaesthetic agent was administered before lumbar puncture.*”

Response 11: Throughout the manuscript, “delirium” refers to collectively preoperative and postoperative delirium, and in analyses were preoperative (prevalent) and postoperative (incident) delirium are studied separately this is specified.

(12) Moderate concern also exists for Tables 2 and 3, which could be simplified by focusing on postoperative delirium and running different models for MMPs/TIMPs while adjusting for dementia. Complex data could be relegated to supplemental materials for clarity.

Response 12: We hope the reviewer will agree that we have sufficiently addressed this comment in our previous response **(2)**. In addition, to simplify the results section we have moved Table 3, showing MMP and TIMP levels in hip fracture patients stratified by delirium and dementia status to the Supplementary material.

(13) A major concern surrounds Table 4, as it forms the core of the paper. Understanding the underlying data, including total sample size and the inclusion of patients with dementia, is essential. Explicitly adjusting for dementia, rather than IQCODE, may be appropriate here. Providing insights into the approach used for data variables chosen as covariates of interest (data-driven, clinical intuition, or both) would enhance the readers' understanding of the results.

Response 13: For clarity we have included the total sample size of the hip fracture cohort ($n = 279$) in the Table 2 (previously Table 4) title. Further we have amended the accompanying note to clarify that we have adjusted for the presence/absence of dementia: “*Adjusted for sex, age, and *presence/absence of dementia* ($IQCODE \geq 3.44 = dementia (n = 118)$, $IQCODE < 3.44 = no dementia (n = 161)$). In all models, higher age and *presence of dementia* were significant predictors of delirium.” We hope that these changes, together with the changes to the methods section detailed previously enhance the understanding of the underlying data.

REVIEWERS' COMMENTS:

Reviewer #1 (Remarks to the Author):

The authors have appropriately addressed the raised concerns. No further comments.

Reviewer #2 (Remarks to the Author):

The authors have addressed satisfactorily all my comments and concerns.

Author responses to reviewer comments:

We would like to thank both reviewers for taking the time to review our manuscript again and acknowledge the revisions we have made. The insightful comments provided in the peer review process has contributed to improving our manuscript.